# Food Informatics—Review of the Current State-of-the-Art, Revised Definition, and Classification into the Research Landscape

**DOI:** 10.3390/foods10112889

**Published:** 2021-11-22

**Authors:** Christian Krupitzer, Anthony Stein

**Affiliations:** 1Department of Food Informatics, University of Hohenheim, 70599 Stuttgart, Germany; 2Computational Science Lab, University of Hohenheim, 70599 Stuttgart, Germany; anthony.stein@uni-hohenheim.de; 3Department of Artificial Intelligence in Agricultural Engineering, University of Hohenheim, 70599 Stuttgart, Germany

**Keywords:** Food Informatics, Internet of Things, Precision Agriculture, smart agriculture, Internet of Food, Food Computing, Food Supply Chain Management

## Abstract

*Background:* The increasing population of humans, changing food consumption behavior, as well as the recent developments in the awareness for food sustainability, lead to new challenges for the production of food. Advances in the Internet of Things (IoT) and Artificial Intelligence (AI) technology, including Machine Learning and data analytics, might help to account for these challenges. *Scope and Approach:* Several research perspectives, among them Precision Agriculture, Industrial IoT, Internet of Food, or Smart Health, already provide new opportunities through digitalization. In this paper, we review the current state-of-the-art of the mentioned concepts. An additional concept is Food Informatics, which so far is mostly recognized as a mainly data-driven approach to support the production of food. In this review paper, we propose and discuss a new perspective for the concept of Food Informatics as a supportive discipline that subsumes the incorporation of information technology, mainly IoT and AI, in order to support the variety of aspects tangent to the food production process and delineate it from other, existing research streams in the domain. *Key Findings and Conclusions:* Many different concepts related to the digitalization in food science overlap. Further, Food Informatics is vaguely defined. In this paper, we provide a clear definition of Food Informatics and delineate it from related concepts. We corroborate our new perspective on Food Informatics by presenting several case studies about how it can support the food production as well as the intermediate steps until its consumption, and further describe its integration with related concepts.

## 1. Introduction

Scientists have been alerting the world about climate change for a very long time, such as the *World Scientists’ Warning to Humanity* from 1992 and the more recent *World Scientists’ Warning to Humanity: A Second Notice* in 2017. However, it required Greta Thunberg and *Fridays for Future* to raise the awareness about climate change and the necessity to protect the environment and society. One aspect that, on the one hand, impacts climate change but on the other hand is also highly influenced by it, is the production of food. Roughly 11% of the Earth’s population was unable to meet their dietary energy requirements between 2014 and 2016, representing approximately 795 million people [1]. On the contrary, the food production for the population of industrial nations, especially, highly contributes to climate change due to a meat-focused diet, with the expectation of seasonal fruits throughout the entire year as well as a high waste of food [2]. Both situations will become more complex in the next decades as the global population is predicted to grow to 10 billion by 2050 according to the United Nations [1]. This might increase the number of people with insufficiently satisfied dietary energy requirements. The increasing welfare in emerging countries will lead to more people adopting the resource-demanding nutrition of the industry nations.

Traditional food production approaches will not be able to deal with those issues sufficiently; hence, novel approaches are required. Especially the integration of current research advances in the Internet of Things (IoT) seems to be promising in supporting various aspects of food production including farming, supply chain management, processing, or demand estimation. Whereas a commonly accepted definition of IoT is not present in the literature, it is agreed that IoT refers to connected computational resources and sensors which often supplement everyday objects. The sensors support the collection of data which can be analyzed for identifying changes in the environment and the IoT system can react to accommodating those changes. Procedures from Artificial Intelligence (AI)—the idea that machines should be able to carry out tasks in a smart way—and Machine Learning (ML)—techniques for machines to learn from data— can complement the analysis and system controlling process in IoT systems. The actions of analyzing and controlling the IoT systems are also named as a reason for adaptation [3]. The purposeful application of those methods can complement and optimize the existing processes. The research in this field is distributed across several domains, such as Precision Agriculture, Smart Farming, Internet of Food, Food Supply Chain Management, Food Authentication, Industrial IoT (IIoT)/Industry 4.0 for food production, Food Safety, Food Computing, or Smart/Pervasive Health. Often, those concepts overlap and are not completely distinguished.

Another research stream can be recognized under the notion of Food Informatics, which is often understood as data-centric research for supporting food production and consumption (e.g., [4,5,6,7]). However, research alone does not provide a clear concept of Food Informatics. In this review paper, we want to distinguish the various research streams related to the topics of food production and consumption. Further, we motivate our perspective on Food Informatics as a supportive research stream that can contribute to the wide field of applying IoT and AI/ML to optimize food production and, hence, can be seen as an underlying technological basement for the other ICT-related research streams that target aspects of the food supply chain. Additionally, we present several case studies related to the production of food, discuss how Food Informatics contributes to those applications, and highlight the relation to the other presented research streams. In summary, our contributions are threefold:**Delineation of concepts:** We provide a delineation of various concepts related to the digitalization in the food science production;**Definition of Food Informatics:** We review the state-of-the-art in Food Informatics and motivate a new understanding of Food Informatics as supportive discipline for food production and underlying technical basement for digitalization;**Application:** We discuss the potential of IoT and AI/ML to support the process of food production and supply—in our understanding, the central role of Food Informatics—with regard to the socio-technical perspective of the various stakeholders.

However, we do not aim at providing a fully-fledged survey as this would be not possible for a broad coverage of topics. Accordingly, we target providing a systematic mapping [8] approach to offer a cross section of the research landscape. The remainder of this paper is structured as follows: Section 2 compares research streams related to the production and consumption of food. Subsequently, Section 3 presents a new definition of *Food Informatics*. Then, Section 4 presents several research perspectives as well as research challenges when applying information and communication technology (ICT) in the food production domain. Section 5 discusses possible threats to the validity of our claims. Finally, Section 6 discusses related surveys before Section 7 closes this paper.

## 2. Delineation of Concepts

The production of food is a highly complex process. On the one hand, there is a high diversity in the combination of ingredients and intermediaries with many dependencies, for example, in the order of processing. Further, by-products, side-products, or co-products might arise, such as butter milk when producing butter to mention just one example. On the other hand, food has hygienic, olfactory, sensory, or preserving requirements. In general, the food production process can be divided into several phases:**Agriculture**: Production of ingredients/food.**Logistics**: Transportation of food while obeying hygienic constraints.**Processing**: Processing of ingredients to food products in an industrial process.**Retail**: Selling of food.**Consumption**: Humans consume the food.**Food Waste Handling**: Intelligent forms of handling food waste and disposal improves the sustainability (not in the scope of this paper).

In this paper, we see this process as a sequential process. However, in practice, a circular economy might be favorable from a sustainability viewpoint. Further, the mentioned by-products, side-products, or co-products create a value-added network rather than a traditional value chain. However, in this paper we focus on how to support the different steps by ICT. Consequently, a sequential view on the food production will not limit the validity of our arguments.

As a seizable example, we show the different phases of the process for the production of Spätzle, a German pasta (see Figure 1) The production starts with the planting and harvesting of wheat (crop cultivation) as well as the production of eggs (livestock production). Both ingredients are transported to the production facility, where the Spätzle are produced by adding water and salt. Subsequently, the product is delivered to wholesale trades, food retail markets, or directly to the consumer/restaurants, where the product is eventually consumed. In all phases, IoT devices can be integrated to either support data collection or actively control the processes through adaptation, that is, adjust the production process to handle machine faults or use traffic forecasts to re-calculate routes as well as react by adjusting production plans to the delay. Additionally, technology known from Smart Health research, such as wearables, can help to observe the consumption behaviour of consumers. The data collection and analysis is supported by *Edge* and *Cloud* technology. With Cloud resources, we refer to flexible server resources that can be used to complement the often limited computational resources of production machines. Those can be company-internal resources, shared by multiple factories, or external resources offered by Cloud providers such as the Google Cloud Platform, Amazon EC 2, or Microsoft Azure. Edge devices are additional computational resources within a factory that extend the computational resources of production machines.

Several concepts apply methods and technology from computer science, mainly from IoT and AI/ML, in order to support the food production process. Those concepts often address only one phase of the production process. In the following, we discuss and compare the different concepts. The purpose of this section is a delineation of the different research streams rather than a detailed review of each of them.

### 2.1. Precision Agriculture

Clearly, the first step in the food supply chain is comprised by the cultivation of crops, husbandry of livestock, and the overall management of farmland. Besides the actual operations and business aspects, which is usually summarized by the term *farming*, the—from our point of view—more general notion of *agriculture* refers to all the tangent scientific and technological aspirations around it. We therefore use the notion of agriculture as an umbrella term in this article.

The presence of variability and uncertainty inherent in many facets of agriculture has been recognized quite a number of decades ago [9]. With this increasing awareness and a focus on the “field” (in the sense of farmland)—that is, recognizing that, for instance, soil and crop might exhibit varying conditions—combined with technological innovations such as global positioning systems (GPS), microcomputers with increasing computational capacity as well as the advent of autonomous systems/robotics into agricultural machinery, a subarea of agricultural sciences—namely *Precision Agriculture*—can be defined. With the focus on the cultivation land in mind, Gebbers and Adamchuk [10] provide a concise definition of the term *Precision Agriculture* as
“[...] a way to apply the right treatment in the right place at the right time.”

They further specify and summarize the goals of Precision Agriculture as three-fold: (1) The optimization of required resources, for example, the utilized amount of seeds and fertilizers, for obtaining at least the same amount and quality of crops in a more sustainable manner; (2) The alleviation of negative environmental impacts; and (3) improvements regarding the work environments and social aspects of farming in general. An alternative, from the authors’ point of view, very intuitive definition is provided by Sundmaeker et al. [11]. They describe the field of Precision Agriculture as
“[...] the very precise monitoring, control and treatment of animals, crops or $m^{2}$ of land in order to manage spatial and temporal variability of soil, crop and animal factors.”

### 2.2. Smart Agriculture

The advances in ICT—such as smart devices, Cloud and Edge Computing, near field communication (NFC)—observable over the last decades, as well as the resulting technological possibilities in nearly any branch of society and industry—summarized by the term IoT as will be introduced below—naturally also opens a wide variety of adoption scenarios for agriculture. *Smart Agriculture* appears as the most common notion in that respect.

Wolfert et al. [12] review the application of big data in the context of Smart Farming. The survey further provides another concise definition of the term:

“Smart Farming is a development that emphasizes the use of information and communication technology in the cyber-physical farm management cycle.”

As can be recognized, a new term has been introduced in the above definition: *cyber-physical farm*. As is often the case when new technologies are emerging, a variety of terms referring to the essentially same thing appear in the literature. Terms that also show up sometimes include: “Digital Farming”. For the sake of completeness, we want to highlight that the notion *Digital Farming/Agriculture* is sometimes also conveyed to mean the integrated and combined utilization of both precision and smart agriculture concepts. The interested reader is referred to a recent position paper of the Deutsche Landwirtschafts Gesellschaft (DLG) (engl. German Agricultural Society) [13]. Since in this article the spotlight is set on the notion of Food Informatics and not on smart agriculture alone, we proceed without a further differentiation), “e-Farming” or the German term “Landwirtschaft (engl. Farming) 4.0” (the latter intended to relate to the German-coined notion of Industry 4.0). Throughout this work, we only carry the differentiation between Precision Agriculture and smart agriculture for the sake of simplicity.

### 2.3. Industry 4.0/Industrial IoT

The vision of *Industry 4.0* is to integrate the cyber space and the physical world through the digitization of production facilities and industrial products [14]. This synchronizes the physical world and a digital model of it, the so called digital twin. The *Industrial Internet*, also known as *Industrial Internet of Things* (IIoT), enables a flexible process control of an entire plant [15]. The current interpretation of the term appeared with the rise of Cloud technologies. The central elements of both concepts—besides the digital twin—are the smart factory, cyber-physical production systems as well an intelligent and fast communication infrastructure.

The food production may benefit from Industry 4.0 approaches. Predictive maintenance can lead to production increase, especially as machine defects in the context of food production have a more serious impact due to the perishability of ingredients in contrast to tangible product elements in the production area. Further, the flexibility of Industry 4.0 approaches can help to facilitate the production of individual, customized food articles. Luque *et al.* review the state-of-the-art of applying Industry 4.0 technology for the food sector and propose a framework for implementing Industry 4.0 for food production centered around the activities of the supply chain [16].

### 2.4. Internet of Food

The term *Internet of Food* was first used by Kouma and Liu [17]. They proposed to equip food items with IP-like identifiers for continuous monitoring them using technology known from the IoT. Hence, it is a combination of identifiers, hardware, and software to monitor food and allow an observation of the consumers for optimizing nutrition. Somewhat contrarily, other authors describe the use of IoT for food-related purposes rather than the identification aspect as the Internet of Food; an example being smart refrigerators [18]. Holden et al. [19] review current developments in the area of the Internet of Food with a focus on the support of sustainability.

### 2.5. Food Computing

Min et al. [20] present a definition of the term *Food Computing* in combination with a review of the current state-of-the-art. According to them, Food Computing is concerned with the acquisition and analysis of food-related data from various sources focusing on the perception, recognition, retrieval, recommendation and monitoring of food. Hence, Food Computing is a consumer-focused research stream with the objective to support the consumer with respect to optimal nutrition. Data sources can include pictures taken with smartphones, and data from web sites or social media data. Accordingly, the research integrates approaches from information retrieval, picture recognition and recommendation systems as well as prediction. For further information on the relevant approaches, the interested reader is referred to overviews on the current state-of-the-art (e.g., [20,21,22,23]).

### 2.6. Smart Health/Pervasive Health

According to Varshney [24], Pervasive Healthcare can be defined as
“[...] healthcare to anyone, anytime, and anywhere by removing locational, time and other restraints while increasing both the coverage and the quality of healthcare”.

In a similar fashion, authors define the research for Smart Health or Mobile Health [25]. Applications in those areas include health monitoring, intelligent emergency management systems, smart data access and analysis, and ubiquitous mobile telemedicine. Often, those applications rely on wearables—that is, small devices with sensors attached to the body of users—for data collection and signaling of critical health conditions. This requires efficient communication technology, smart IoT devices, and intelligent data analytics. Nutrition monitoring might be a relevant aspect in the health monitoring as well as telemedicine. Vice versa, Smart Health apps might influence the consumption of food [26]. Additionally, somehow related to the this area are newer works that target the field of (personalized) nutrition, for example, smart food choices that support the choice for food of a consumer [27] as well as nutrition informatics, which “describes approaches to understand the interactions between an organism and its nutritional environment via bioinformatics-based integration of nutrition study data sets” [28].

### 2.7. Food Supply/Logistics

Supply chain management describes the optimization of the intra and extra logistics. In the case of food production, this includes the transportation of ingredients to the production facility, the moving of ingredients and products in the facility as well as the transportation to retailers or customers. In contrast to other tangible goods, food has specific requirements concerning the temperature, hygienic aspects, and its storage, for example, avoiding pressure on the products. In the following, we focus on the extra logistics of food, that is, its transportation outside of a production facility. Current approaches try to integrate IoT technology for monitoring of the logistics, especially the monitoring of the temperature and air quality. The application of RFID improves the tracking of food and furthers the information handling [29]. Currently, approaches propose to integrate Blockchain technology into the food supply chain to guarantee traceability [30,31], that is, food provenance. Introini et al. [32] provides an overview on the traceability in the food supply chain.

### 2.8. Food Safety/Food Authentication

According to a recent overview by Danezis et al. [33],
“[...] food authentication is the process that verifies that a food is in compliance with its label description”.

Food Authentication is one part of the Food Safety area, which comprises the monitoring and control of food to guarantee its quality throughout the value chain. Some authors present works that integrate IoT technology, mainly based on sensors for monitoring (e.g., [34,35] to achieve food safety). Recent approaches propose integrating Blockchain technologies to achieve a high reliability and availability of information [30,31]. This might help to increase the security of the stored information; however, one common issue for data-related analysis, the “Garbage In, Garbage Out” principle—which says that the quality of the output of an analysis is determined by the quality of the input—is not solved by the Blockchain technology as it just acts as secured data storage.

### 2.9. Summary

The presented concepts share some similarities. First, the presented approaches can be grouped along the mentioned phases of the food production process: agriculture, logistics, production, and consumption. For retailing, we focus on the logistics part. Hence, we did not explicitly discuss retailing specifics. Precision and smart agriculture is mainly concerned with the operational (and scientific) aspects of crop and livestock production as well as farmland husbandry and management. IIoT and Internet of Food approaches concentrate on supporting the production of food. The consumer-centering research domains, Smart Health and Food Computing, target the optimization of the food consumption behavior. The logistics aspects of food supply links the different phases of the process. Food Authentication spans the whole process chain as it provides a continual monitoring of food; however, it is limited to the activity of monitoring the process to guarantee the authenticity of the ingredients and products. Accordingly, those concepts provide customized mechanisms for specific tasks; however, they are not generically applicable or reusable in several phases of the food production process.

Second, the presented research streams rely on advances in IoT (mainly on sensors for data collection) and AI (mostly autonomous robotics and ML). However, researchers mostly try to integrate or customize existing technology instead of developing new methodologies optimized for the requirements specific to food production. Furthermore, often the suggested technology is customized to very specific purposes instead of providing more generic and flexible frameworks that can be used in several phases of the entire food production process with only minor adjustments.

Third, some research streams are related. Smart agriculture and Precision Agriculture both address the agricultural process part and can be integrated to maximize their benefits. The Internet of Food research stream overlaps with food supply as it addresses the monitoring of food. Further, as monitoring of food is an inevitable element for the Food Authentication, Internet of Food is also related to Food Authentication and food safety. Lastly, Food Computing and Smart Health overlap in their purpose as well as some methods, for example, data extraction from pictures captured with smartphones.

Consequently, we propose the development of generic approaches relying on IoT and AI that can support various process steps. This seems especially beneficial for data analytics procedures to analyze sensor data or forecast future system states, as those implement generic ML mechanisms. In the next section, we present how Food Informatics could step into the breach by means of proposing a new definition, which comprises our notion of the term.

## 3. A Revised Definition of Food Informatics

A particular research direction from the food-related literature that sets the incorporation of concepts from computer science as an enabling technology in the spotlight is summarized under the notion of *Food Informatics*. As shown in Figure 2, Food Informatics can be vaguely defined by integrating the different perspectives and research streams as delineated above.

The authors of [4] understand and motivate Food Informatics as a mainly data-driven perspective. This includes the development of tools and technologies to enable the application of ontologies for sharing knowledge specific to the food production process [5,6,7]. Similarly, according to some authors [36,37], Food Informatics deals with collecting information and documenting health and medicine related information. On the contrary, the following definition [38] also includes the reaction on the analysis of the collected information while limiting the application to the end users:

“Food informatics is a specific eHealth area for the prevention and management of overweight and obesity.”

Lastly, Martinez-Mayorga and Medina-Franco [39] relate chemoinformatics—the use of computers to collect and manipulate chemical information—to Food Informatics. They define Food Informatics as the application of chemical information to food chemistry. Martinez-Mayorga et al. [40] present an overview of databases and software for chemoinformatics.

The large diversity of definitions demonstrates that the meaning of the term *“Food Informatics”* has not yet converged to a consensus. Still, all definitions at least focus on the data collection and use of the data related to food. However, while some works focus on the food production [4,5,39], others highlight the importance of integrating consumers [36,38]. This shows a large diversification and spans almost the whole process of food production. Additionally, the application of the collected information differs from providing ontologies [4,5], integrating technology for data collection [5], the use of informatics to analyze the collected data and reacting accordingly [36,38], or even integrating other nature science disciplines for information retrieval [39]. Summarizing, no currently available definition for Food Informatics covers all relevant aspects.

The existing definitions target the phases of food production and data management as well as Smart Health. As the production of food is an interplay of many different processes in agriculture, production systems, supply chain management, and Smart Health with obvious interdependencies, we propose to also include the data/information acquisition from the very beginning; hence, during crop and livestock production (smart agriculture), and to also take information collection for logistics and transportation into consideration. We deem a span over the entire process important, as issues in one process step might impact other process steps. For instance, insufficient handling of food during the transportation can negatively impact the food quality for the customers. Accordingly, a holistic information perspective is important. Various technologies can support the collection of such information, especially IoT technology. Furthermore, the analysis of the collected data can highly benefit from (Deep) ML and data analytics techniques. Approaches from the research domains concerned with adaptive systems, for example, self-adaptive systems [3], self-aware computing systems [41], or Organic Computing [42], can support the implementation of mechanisms that allow for adequate reactions according to the analyzed information. A robust self-reconfiguration to react to unexpected events, such as machine defects in the food production facilities, constitutes an example for that. However, due to the hygienic, taste-related, or legal constraints, the area of food production has many domain-specific requirements that must be satisfied. Hence, we propose the customization of computational approaches optimized for the specifics of the food domain. This is exactly what, from our point of view, should be the central task of Food Informatics. To reflect all considerations from above, we therefore suggest a new definition:
**Food Informatics is the collection, preparation, analysis and smart use of data from agriculture, the food supply chain, food processing, retail, and smart (consumer) health for*****knowledge extraction*****to conduct an*****intelligent analysis*****and reveal*****optimizations*****to be applied to*****food production*****,*****food consumption*****, for*****food security*****, and the end of life of food products.**

This new definition stresses the relevance for integrating computer systems and ICT into the food production process. It is related to the other concepts presented in Section 2, as those concepts can be seen as specialized subfields of Food Informatics. The definition covers all aspects of the food production process and can also include relevant aspects from a circular economy perspective. It very much benefits from recent advances in the field of artificial intelligence, as those contributions support the intelligent reasoning, that is, the analysis of current and forecasted system states and situations to optimize the food production processes through adaptations and adjustments. The intelligent and purposeful application of informatics opens a variety of use cases concerning food production and consumption. This can also support the transformation from linear supply chains to a circular economy as the digitization of information supports the analysis of data and the optimization of side streams and the end of life of products, and hence, support to create a feedback loop, that is, circular loop. The next section presents such use cases.

## 4. Food Informatics in Pratice: Today and Tomorrow

As discussed in Section 3, we define Food Informatics as the purposeful application of methods from different areas of computer science to the food production process. This is a rather technology-oriented and also holistic view. However, this is what was intended by us: we claim that Food Informatics provides the underlying technological basement, that is, representing the digitalization of the food industry, and its specific facets can be seen in many different manifestations of scientific concepts (see Section 2) that address specific concerns in the food supply chain. As ICT always includes a socio-technological perspective, this section presents several case studies that show how Food Informatics can support all the consecutive phases of the food supply and how stakeholders interact as well as how Food Informatics is delineated from but also complements the other research streams presented in Section 2. The case studies are ordered “from the field to the customer”, that is, in the chronological order of the production steps. Figure 3 provides an overview of these use cases and integrates them along the food production chain. In the following, we explain each case study in detail and describe how Food Informatics can contribute to the use cases and discuss how it is related to the research streams presented in Section 2.

### 4.1. Autonomous Robotics in Precision Agriculture

As we already defined in Section 2, Precision Agriculture is concerned with handling the spatial and temporal variability inherent in many facets of agricultural processes. For instance, autonomous land machines or robots are utilized to monitor soil quality via the attached soil sampling equipment (sensors) and precisely apply a site-specific amount of fertilizers to compensate for nutrient-deficiency. This methodology is called *Variable Rate Nutrient Application* (VRNA). Here, AI methodology can be applied to infer so-called prescription maps with the most effective and cost-efficient soil-sampling schemes, as presented by Israeli et al. [43].

Needless to say, cost-efficiency plays a central role when creating such field mappings to predict crop yield or make use of VRNA. According to Boubin et al. [44], computation costs for frequent yield mappings might consume a large fraction of the profits obtained by the farmers for crop cultivation. Therefore, fully autonomous aerial systems (FAAS), that is, drones not operated by human pilots, are deemed more cost-efficient. FAAS, however, demand a non-negligible amount of computing resources in order to leverage powerful vision capabilities and AI technology. This is where swarms of drones enter the field, together with Edge to Cloud-based Computing infrastructures [44].

As a collective of FAAS, tasks such as achieving a complete field coverage can be distributed among the swarm. For instance, within the current research project called SAGA, fully autonomous drones operate on different levels of altitude to partition the monitored field into sectors and instruct lower flying drones to inspect the crop sectors for weed or plant diseases [45,46]. The utilization of ensembles of self-integrating heterogeneous autonomous/robotic systems, where FAAS collaborate with mobile ground robots equipped with sensors and actuators, for example, for precise weed treatment or fertilizer application, bears great potential for modern Precision Agriculture, but also presents technological challenges that need to be overcome [47].

In the context of Food Informatics, as depicted in Section 3, it becomes apparent that access to Food IoT services hosted in the Cloud constitutes a key aspect. As a result, Business Intelligence or other data analytics applications can be leveraged. This leads to potential Food Informatics use cases such as:
Demand-based supply from the input industry (fertilizers, herbicides, pesticides) in line with current field conditions (soil nutrients, plant health) and environment factors (droughts, long winters);Crop condition-aware and treatment-specific adaptive pricing models for wholesale and, in turn, final retail;Exact site-specific crop/livestock treatment information (using GPS or NFC technology) to allow for food traceability “from field to fork”.

Furthermore, the deployed swarm robots or autonomous land machines can be equipped/ retrofitted with special-purpose sensors to continually monitor their system-health status. Using the acquired data, predictive services can adequately plan maintenance works and consequently reduce down times and, therefore, possible yield losses or food waste.

### 4.2. AI/ML-Supported Smart Agriculture

The rise of AI technology and especially deep learning solutions—mainly the increasing amount of available *big data* and continually progressing advances in high-performance computation capabilities for their processing [11]—offer various potentials for the application of ML to agriculture. Recent surveys on the use of (Deep) ML applications for smart agriculture can be found (e.g., [48,49]).

Wahby et al. [50] present an intriguing example of ML applied in a smart gardening scenario, which appears seamlessly adoptable to crop plant growth in the agricultural context. They train an ML model based on recurrent LSTM networks which predicts the underlying plant growth dynamics, that is, the stiffening and motion behavior of a bean plant as a response to controllable light stimuli. This model is subsequently used to evolve a controller for an entire bio-hybrid setup, which allows the modification of the plant’s growing behavior by exploiting the phototropism property. Such sensor-actor (robotic) systems will attract more attention in the future and will prove crucial for robust indoor-cultivation of crops in urban areas (*urban/indoor farming*). Further, applications of Organic Computing [42] target livestock management [51] and autonomous off-highway machines [52].

Since AI and ML both constitute two of the most investigated subfields of computer science these days, they clearly also play a central role in smart agriculture and, thus, in Food Informatics. Scenarios are imaginable where urban greenhouses, equipped with self-adaptive bio-hybrid systems (as delineated above), support a sustainable and robust crop cultivation regardless of the season and current weather conditions in order to ensure food security. Connected to Cloud and IoT services, demand and weather forecasts can be incorporated to approach intelligent food production systems that are more cost-effective and at the same time minimize food waste while still satisfying current needs. This would allow, for example, for site-specific productions of crops on-demand which bears the potential of reducing logistic costs and pollution.

### 4.3. Internet of Things and Blockchain-Supported Food Supply

The food supply chain integrates all process steps and supports a continuous tracking of the food throughout the production process. Hence, many parties work together. Such a cooperation requires reliable data exchange. However, a central shared data repository constitutes a single point of failure as well as a potential performance bottleneck. Further, the diversity of actors triggers the question about where to establish such a central data repository. Accordingly, distributed data management solutions might be beneficial, as those reduce data duplication and increase the robustness of the data access. *Carrefour* is among the first industry companies relying on the Blockchain technology for the purpose of food supply chain data management. However, so far the roll-out of this technology is limited and mainly serves as an experimental marketing use case for a specific product. Several authors (e.g., [30,31]) propose to integrate the Blockchain for traceability purposes, as the complete documentation of the origin of ingredients and food is highly important and often a legal obligation. Kamilaris et al. [53] provide an overview of the use of blockchains in the agri-food supply chain.

A key task in the food supply chain is the logistics. Contrary to the logistics of common products, food entails several requirements due to its perishability. This includes cooling, hygienic constraints, or avoiding pressure on the surface of food. RFID and NFC technology might support the traceability of the items [35]. IoT technology, mainly intelligent sensors, can improve the monitoring of the conditions during the transportation of goods [29]. Further, ML-supported analysis of data can help to optimize the process, for example, by forecasting the arrival of items in the production facility and, thus, reducing delays regarding subsequent processing steps.

Food Informatics can contribute on several ways. The definition of common data description and knowledge representation formats, for example, in the form of ontologies [5,6,7]. Further, it can support the data exchange with generic services to store and access data in the Cloud or the Blockchain. Additional services can offer generic interfaces to store data sensed by IoT devices into the shared data storage or generic tools for ML-supported data analytics. Such services will further contribute to various activities in the food supply chain.

### 4.4. Items-Focused Data Collection in Food Production

Industry 4.0 and IIoT approaches promise a flexible production by means of collecting and analyzing data. The reconsideration that a product itself should determine its production steps rather than the processing machines constitutes one key aspect for instance. Therefore, Industry 4.0 and IIoT approaches integrate intelligent data analytics. So far, the collection of the required data mainly focuses on the state of machines or the quality of the intermediate or final products w.r.t. pre-defined quality ranges. However, for a detailed analysis of products’ quality issues the collection of machine data alone might not be sufficient to identify production issues; this also requires the collection of product-related data.

Maaß, Pier and Moser [54] describe the design of a smart potato. Using IoT technology and sensors, a dummy potato can deliver information from the harvesting process, for example, the pressure of the harvesting machine on the potatoes. In several studies, the authors captured the effects of different acceleration patterns on the skin of a potato. Using these data, they trained deep learning algorithms to automatically analyze whether the pressure of a harvesting machine can damage a potato.

Such an approach might be plausibly transferred to the food production. Using IoT dummy food items throughout the production in order to collect data from the products’ viewpoints can complement the purely machine-centered data. With this food item related data perspective, quality issues such as too much exerted pressure on the ingredients can be straightforwardly identified. Again, Food Informatics can contribute with generic data collection based on sensors from the IoT and ML-driven data analytics services.

### 4.5. An Adaptive, Flexible Food Production

One of the main objectives for Industry 4.0 and IIoT is to provide a flexible production that supports the individualization of products [15,55,56]. Examples are cars, furniture (such as tables or cabinets), or personalized books. Consequently, a targeted lot size of 1 requires a flexible product design as well as an adaptive production process.

A recent study in the German food industry [57] identified that two third of the companies pursue a lot size of 1 by 2030. Hence, it seems beneficial to integrate mechanisms known from the areas of self-adaptive systems [3], self-aware computing systems [41], or Organic Computing [42] to support a flexible, robust and adaptive food production. Further, such a robust adaptive production process is able to tolerate fluctuations in the quality/size of the ingredients.

Food Informatics can provide a powerful framework for supporting the adaptivity of intelligent production systems which are customized to the specifics of the food industry. Furthermore, it can support the integration of emerging technologies that can foster the individualization of food items, such as additive manufacturing via 3D printers [58].

### 4.6. Predictive Maintenance in the Food Production

Predictive maintenance is based on the idea that certain characteristics of machinery can be monitored and the gathered data can be used to derive an estimation about the remaining useful lifetime of the equipment [59]. This can help to predict potential machine defects in advance and reduce or even eliminate delays in the production process as a result of machine defects and downtimes. The underlying problem hereby is the detection of anomalies in the machine data [60].

Although it is clearly understood that such production delays imply monetary losses in the production of normal goods, the consequences of such unexpected production downtimes are even worse for the production of food due to its perishability. Accordingly, the utilized prediction and forecasting methodologies demand for customized algorithms and, thus, advanced development and domain knowledge.

Recommendation systems (such as [61]) can aid the process of automatic identification of the most adequate forecasting algorithm fitting the underlying data patterns. The selection of the most appropriate algorithm might then be combined with automatic algorithm configuration or hyperparameter tuning [62] for optimizing the parameter setting of the algorithm to be utilized. Food Informatics should contribute here by means of conducting research in both areas. That is, to provide predictive maintenance automatically optimized to the specific requirements of food production, for example, by focusing on forecasts of machine defects with time horizons that consider the foods’ perishability and cooling requirements. Further, those recommendation systems can be re-used for other forecasts, for example, forecasting the transportation time or the demand for specific food.

### 4.7. Demand-Driven Food Production

For particular industries, it is common to start the production just after an incoming order, for example, for cars. This reduces the likelihood of overproduction but on the other hand results in waiting time for customers. For the case of food, such a policy bears additional benefits due to the perishability of the produced food items. Additionally, such forecasts help to identify trends early: given the time required from planting ingredients to the final products, the forecasts help to change the supply chain early in advance to accommodate the trends.

A sensible trade-off between a production in stock, as well as a purely demand-driven production, could be the integration of demand forecasting by identifying food consumption trends. Research streams as Food Computing [20] and Smart Health [26] can contribute to the analysis of consumption behaviors and forecasting of food demands due to their methods for information extraction. Embedding such demand forecasts into a feedback loop can optimize the various aspects from the food production to the consumption behavior and eventually reduce food waste. Coupled with adaptive food production systems as outlined above, this constitutes a promising way for achieving sustainable food chains.

Food Informatics can contribute by offering services of knowledge extraction regarding food trends, for example, from social media and Smart Health technology. This can be combined with powerful data analytics and forecasting techniques, such as the already proposed forecasting recommendation systems for choosing the prediction algorithms.

## 5. Threats to Validity

In this paper, we target providing a systematic mapping [8] approach to offer a cross section of the research landscape. Consequently, we do not follow a systematic approach to identify all relevant works for an area. On the one hand, this is hardly feasible. Our aim is to provide an overview paper on the application of ICT on the agri-food industry. This is such a broad field, so that it is just impossible to cover each facet in detail. On the other hand, this is not our intention; we want to focus on the application of the term “food informatics” and position this concept in the research landscape.

We omit in this paper a detailed analysis of the identified approaches. Again, this is not our purpose; we rather want to span the scope of the research landscape. Accordingly, we do not analyze approaches in detail. Several other surveys with a more narrow scope provide those information (see Section 6).

Instead of providing a fully-fledged survey, we aim to present an overview including a broad coverage of topics. Still, it is feasible that we miss topic. Further, at some point we had to limit the granularity of topics, for example, when talking about food safety it would also be possible to cover the related topics’ shelf-life prediction of HACCP or food logistic might include topics as cold chain and live animal transportation. Again, as we do not want to go into detail, we had to cut at some point and narrow our analysis for the covered topics.

## 6. Related Work

Several surveys and overview articles focus on one of the presented research areas. Min et al. [20] review approaches from information retrieval, picture recognition and recommendation systems as well as prediction for their applicability in Food Computing. Zhong et al. [63] discuss and compare systems and implementations for managing the food supply chain. Verdouw et al. [64] and Tzounis et al. [65] review systems and challenges for supporting agriculture with IoT. [12] emphasize the chances for integrating Big Data concepts for analyzing agricultural processes. Holden et al. [19] review approaches for the Internet of Food and discuss how those contribute to sustainability. However, none of the aforementioned reviews target several aspects of the food production to consumption chain as is deemed essential in our perspective on Food Informatics.

Other review articles focusing on IoT/IIoT present the application of those topics in the food industry. Al-Fuqaha et al. [66] present an overview on technologies and protocols for the IoT and discuss their applicability in an eating order use case. Similarly, Javed et al. [67] and Triantafyllou et al. [68] review recent IoT technology and describe its application in the context of smart agriculture. Xu et al. [15], Sisinni et al. [55] and Liao et al. [56] review approaches for the IIoT and explicitly describe how to adopt them for food production. Ben-Daya et al. [69] review supply chain management approaches and identified that many approaches target the delivery supply chain process and the food supply chains. Food production constitutes one among further aspects in all of those overviews, but is not treated as the central issue there. Further, those works focus on only one aspect of the food production process.

## 7. Conclusions

The production and consumption of food highly benefits from the application of IoT and AI technology. This can especially reduce the waste of food by optimizing the production according to the customer demands. So far, various research streams focus on different aspects of the production process. However, they miss the methods and approaches that can be applied across several steps along the food production process. Further, they often use generic IoT technology and data analytics methods rather than devising methods that are customized for the food production sector. Consequently, we propose to extend the often data-driven perspective on Food Informatics to a generic ICT-fueled perspective, which comprises the application of ICT—mainly IoT and AI/ML—in order to optimize the various aspects and processes concerning food production, consumption and security.

This paper provides a motivation and revised definition for Food Informatics and corroborates our perspective with potential use cases. As next steps, we will provide a comprehensive framework based on the revised definition and the envisaged applications. Furthermore, we will present how to adopt existing IoT and AI-based procedures and tools, and subsequently demonstrate their applicability in use cases of digital farming (i.e., precision and smart agriculture) and the processing of food in the context of Industry 4.0. Additionally, in this paper we focus the traditional economy model. For future work, we plan to further elaborate the application of food informatics to support the transition towards a circular economy and also extend the perspective towards the bio-based industry beyond food products.

## Figures and Tables

**Figure 1 foods-10-02889-f001:**
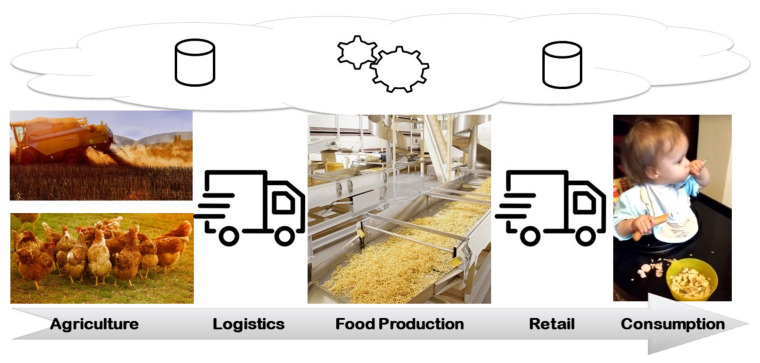
Overview on the different activities in the food supply chain using the example of Spätzle production.

**Figure 2 foods-10-02889-f002:**
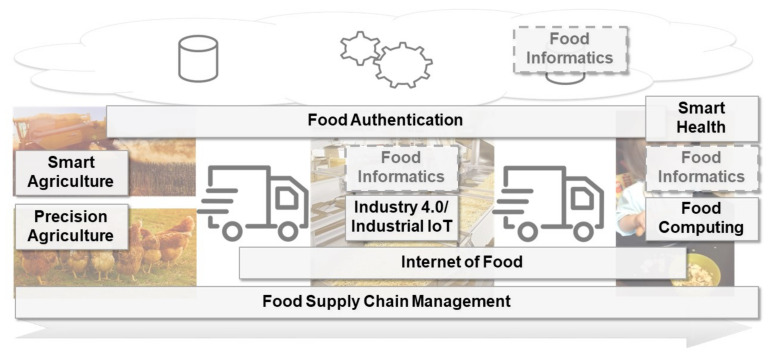
Presented scientifc concepts mapped to the food supply chain.

**Figure 3 foods-10-02889-f003:**
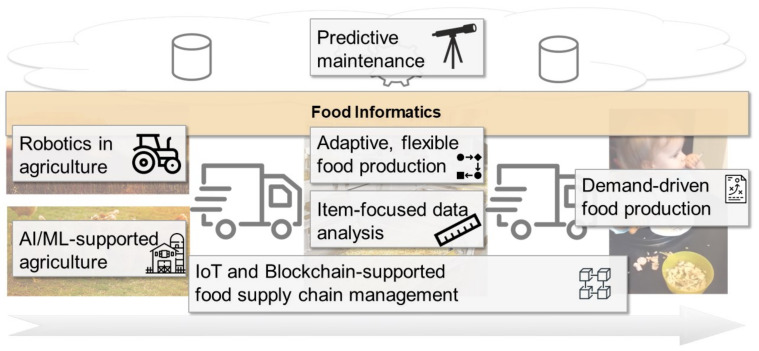
Landscape of use cases mapped to the food supply chain.

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
