# Peer review of "Food Informatics—Review of the Current State-of-the-Art, Revised Definition, and Classification into the Research Landscape"

_foods, 2021, doi:10.3390/foods10112889_

Round 1

Reviewer 1 Report

In this paper the authors introduce as an inventive step with respect to the prior art the definition of Food Informatics. However they stated in the conclusions: "For future work, we plan to integrate a circular economy perspective and also the bio-based industry". Indeed in the definition made in section 3 there is no trace of the end of life of food production and consumption and in my opinion the definition should be integrated with this aspect to comply the need to integrate circular economy concepts and to make the paper coherent in all the sections it has. The definition of Food Informatics should contain also the end of life of foods.

line 358-359 "Please punctuate equations as regular text. Theorem-type environments (including
359 propositions, lemmas, corollaries etc.) can be formatted as follows: "

please delete notes from the paper

line 361-362: it seems from the definition made in section 3 that Food Informatics cover the entire life cycle of a food and not only food production. The statement should be be coherent with the above mentioned definition.

Author Response

In this paper the authors introduce as an inventive step with respect to the prior art the definition of Food Informatics. However they stated in the conclusions: "For future work, we plan to integrate a circular economy perspective and also the bio-based industry". Indeed in the definition made in section 3 there is no trace of the end of life of food production and consumption and in my opinion the definition should be integrated with this aspect to comply the need to integrate circular economy concepts and to make the paper coherent in all the sections it has. The definition of Food Informatics should contain also the end of life of foods.

Answer: We thank the reviewer for pointing to this issue. We revised the paper to better cope the full life cycle of the supply chain.

line 358-359 "Please punctuate equations as regular text. Theorem-type environments (including
359 propositions, lemmas, corollaries etc.) can be formatted as follows: "

please delete notes from the paper

Answer: We deleted this and apologize for overlooking it.

line 361-362: it seems from the definition made in section 3 that Food Informatics cover the entire life cycle of a food and not only food production. The statement should be be coherent with the above mentioned definition.

Answer: Indeed, this seems a logical misfit. We revised the paper to better cope the full life cycle of the supply chain.

Reviewer 2 Report

The authors aim at defining the concept of Food Informatics by firstly surveying how it is used and defined in the literature, and then proposing a definition on their own. 

Direct to the point, I do not find the definition to be compelling. Food Informatics seems to be a wide umbrella term comprising every use of ICTs in all steps of (what is typically referred to as) the food chain (although I agree that a circular view - maybe a grid - would be more truthful with respect to reality). What seems to be missing is the socio-economic perspective and the impacts due to the use of ICTs in the agrifood field. Considering for instance Fig. 3, it seems that all phases (and all use cases) fall into Food Informatics, which makes the definition too wide and too vague, in my opinion. I would clarify, for instance through examples, what should not be considered as Food Informatics for the sake of clarity.

The paper needs a careful review to remove lots of typos, statements coming from the template and not deleted ('Please punctuate equations as regular text. Theorem-type environments (including 359 propositions, lemmas, corollaries etc.) can be formatted as follows:'), and so on. 

Author Response

Direct to the point, I do not find the definition to be compelling. Food Informatics seems to be a wide umbrella term comprising every use of ICTs in all steps of (what is typically referred to as) the food chain (although I agree that a circular view - maybe a grid - would be more truthful with respect to reality).

Answer: We acknowledge that our definition provides a rather technology-oriented and also holistic view. However, this is what was intended by us: We claim that Food Informatics provides the underlying technological basement, i.e. representing the digitalization of the food industry, and its specific facets can be seen in many different manifestations of scientific concepts that address specific concerns in the food supply chain. As it seems that the paper did not transport this sufficiently to the reader, we revised the paper at several locations.

What seems to be missing is the socio-economic perspective and the impacts due to the use of ICTs in the agrifood field.

Answer: Thanks for pointing out this issue. We added a comment on this issue in Sect. 4.

Considering for instance Fig. 3, it seems that all phases (and all use cases) fall into Food Informatics, which makes the definition too wide and too vague, in my opinion. I would clarify, for instance through examples, what should not be considered as Food Informatics for the sake of clarity.

Answer: Please see our comment above to the first reviewer comment. We tried to make it in the paper more clear 

The paper needs a careful review to remove lots of typos,

Answer: We did another round of proofreading to correct those issues.

statements coming from the template and not deleted ('Please punctuate equations as regular text. Theorem-type environments (including 359 propositions, lemmas, corollaries etc.) can be formatted as follows:'), and so on. 

Answer: We deleted this and apologize for overlooking it.

Thanks to the reviewer for investing time to improve the paper's quality. We appreciate the comments. 

Round 2

Reviewer 1 Report

The authors have correctly addressed alla the comments I made in the first revision, so in its present form is worth of publication.

Reviewer 2 Report

I thank the authors for addressing my comments and making their claim more clear. I see that now the limitations - if one could call them so - of the validity of the proposed definition are stated in Section 4.

some minor comments:
line 300: I would suggest opening with 'Reference [4]' rather than just '[4]
line 373: I would rephrase in 'As ICT has the potential to trigger changes in a socio-technical system'
line 404: remove the double '.'